# Nuclear organisation and replication timing are coupled through RIF1–PP1 interaction

Stefano Gnan [1,2,7], Ilya M. Flyamer[3,10], Kyle N. Klein [4,10], Eleonora Castelli[2,8], Alexander Rapp [5], Andreas Maiser [6], Naiming Chen [2], Patrick Weber[5], Elin Enervald[1,2,9], M. Cristina Cardoso [5], Wendy A. Bickmore [3], David M. Gilbert [4] & Sara C. B. Buonomo [1,2✉]

Three-dimensional genome organisation and replication timing are known to be correlated, however, it remains unknown whether nuclear architecture overall plays an instructive role in the replication-timing programme and, if so, how. Here we demonstrate that RIF1 is a molecular hub that co-regulates both processes. Both nuclear organisation and replication timing depend upon the interaction between RIF1 and PP1. However, whereas nuclear architecture requires the full complement of RIF1 and its interaction with PP1, replication timing is not sensitive to RIF1 dosage. The role of RIF1 in replication timing also extends beyond its interaction with PP1. Availing of this separation-of-function approach, we have therefore identified in RIF1 dual function the molecular bases of the co-dependency of the replication-timing programme and nuclear architecture.

[1] Epigenetics & Neurobiology Unit, European Molecular Biology Laboratory (EMBL Rome), Monterotondo, Italy. [2] Institute of Cell Biology, School of Biological Sciences University of Edinburgh, Edinburgh, UK. [3] MRC Human Genetics Unit, Institute of Genetics and Cancer, University of Edinburgh, Edinburgh, UK. [4] Department of Biological Science, Florida State University, Tallahassee, FL, USA. [5] Cell Biology and Epigenetics, Department of Biology, Technical University of Darmstadt, Darmstadt, Germany. [6] Department of Biology II, LMU Munich, Munich, Germany. [7] Present address: Institut Curie, Université PSL, Sorbonne Université, CNRS UMR3244, Dynamics of Genetic Information, Paris, France. [8] Present address: Friedrich Miescher Institute for Biomedical Research, Basel, Switzerland. [9] Present address: Department of Molecular Biosciences, The Wenner-Gren Institute, Stockholm University, SE-106 91 Stockholm, Sweden. [10] These authors contributed equally: Ilya M. Flyamer, Kyle N. Klein. ✉email: sara.buonomo@ed.ac.uk

n eukaryotes, origins of DNA replication are not activated all at once. Origin firing follows a cell-type specific temporal programme known as DNA replication timing. The replication-timing programme is mirrored by the spatial distribution in the nucleus of replication foci, which are clusters of about five simultaneously activated bidirectional replication forks[1]. Both spatial and temporal replication patterns are re-established every cell cycle in G1, at the timing decision point (TDP)[2], that coincides with chromosomal territories achieving their radial position[3] and the re-establishment of chromatin architecture and interphase-nuclear configuration[4]. The spatial organisation of DNA replication is evident at multiple levels. The units of DNA replication timing, replication domains (RD), coincide with one of the basic units of three-dimensional (3D) genome organisation, the topologically associated domains (TADs)[5]. Recently, in cis elements (early replicating control elements—ERCEs) that can simultaneously influence chromatin looping and replication timing have also been identified[6]. Moreover, the "assignment" of RDs as early or late replicating (the establishment of the replication-timing programme), takes place on a chromosome-domain level, prior to the specification of the active origins of replication[2]. On a global scale, the early and late replicating genomes overlap with the A and B compartments identified by Chromosome Conformation Capture methods (HiC)[7–9] and are segregated in the nuclear interior or the peripheries of the nucleus and nucleolus, respectively. It has been shown that artificially re-localising chromocenters to the nuclear periphery affected their replication timing without an immediate impact on their epigenetic makeup[10]. Finally, a recent study from budding yeast has shown that activation of early origins drives their internalisation[11]. However, no molecular, causal link between the temporal and spatial aspects of DNA replication organisation has been established.

RIF1 is a key genome-wide regulator of replication timing[12–18]. It is also involved in re-establishing spatial chromatin organisation in the nucleus at G1[13], and in the control of replication foci spatial dynamics[12]. RIF1 could therefore be a molecular connection between the temporal and spatial organisation of DNA replication in mammalian cells.

The molecular function of RIF1 is still unclear, although it is involved in a variety of functions such as DNA repair[19–30], telomere length regulation in yeast[31–35], cytokinesis[36], epigenetic[37–41] and DNA replication-timing control. Mammalian RIF1 (266 kDa) interacts with components of the nuclear lamina[13,42], behaving as an integral part of this insoluble nuclear scaffold and chromatin organiser. RIF1 associates with the late replicating genome, forming megabase-long domains called RIF1-associated-domains (RADs)[13]. It is unknown what directs RIF1's association with chromatin, but both the N and C terminus can mediate the interaction with DNA[33,43–47]. RIF1 has a highly conserved interaction with protein phosphatase 1 (PP1) that is reported to be critical to regulate the firing of individual late origins of replication[15,48–51]. Activation of these origins is promoted by RIF1 removal in late S-phase, led by the increasing levels of cyclin-dependent kinase (CDK) activity[15,16,48–51]. These studies therefore place the role of the RIF1–PP1 interaction at the stage of execution of the replication-timing programme, in S-phase. However, we have also identified a role for RIF1 as a chromatin organiser earlier during the cell cycle, in G1, around the time of the establishment of the replication-timing programme[13]. Rif1 deficiency impacts nuclear architecture, relaxing the constraints that normally limit chromatin interactions between domains with the same replication timing[13]. It is unknown if RIF1-dependent chromatin architecture establishment affects the replication-timing programme, how RIF1 contributes to nuclear organisation, and if and how its interaction with PP1 plays a role in these functions. More

generally, the functional relationship between nuclear architecture and replication timing is still unclear.

Here, we tackle these questions by interfering with the RIF1–PP1 interaction, introducing point mutations in Rif1 that specifically abolish the interaction. Our results show that both replication timing and nuclear organisation depend upon RIF1–PP1 interaction. However, unlike the replication-timing programme, we find that nuclear organisation is exquisitely sensitive to RIF1 dosage. Using this separation-of-function approach, we identify in RIF1 the molecular hub for their co-regulation. In addition, we show for the first time that the replication-timing programme can be established and executed independently of a specific 3D organisation or of the spatial distribution of replication foci.

## Results

**Mouse embryonic stem cells (ESCs) expressing $Rif1^{\Delta PP1}$.** RIF1–PP1 interaction promotes the continuous dephosphorylation of MCM4 at replication origins that are "marked" to be activated only during the later part of S-phase[15,48–51]. This suggests that, through RIF1, PP1 contributes to control of the time of firing of individual origins of replication. However, the functional significance of RIF1–PP1 interaction for the establishment and domain-level regulation of the replication-timing programme, and in the context of nuclear 3D organisation is unknown.

Mutations that perturb RIF1–PP1 interaction are potential tools to achieve separation-of-function between nuclear organisation and replication timing. We have recently identified the sites within RIF1 that mediate the physical contacts with PP1 (SILK and RVSF motifs, respectively residues 2128–2131 and 2150–2153). Point mutations of these residues reduce RIF1's interaction with PP1 to undetectable levels (RIF1$^{\Delta PP1}$: SILK into SAAA and RVSF into RVSA[52]). We therefore sought to express the $Rif1^{\Delta PP1}$ mutant allele in mESCs. Rif1 overexpression is toxic, hence, to create a system to expresses $Rif1^{\Delta PP1}$ at controlled and as physiological levels as possible, we have utilised $Rif1^{FH/flox}$ mESCs. In these cells, one allele of Rif1 contains loxP sites flanking exons 5 to 7[19] ($Rif1^{flox}$), while the second is a knock-in of a FLAG-HA2 tag (FH) into the Rif1 locus ($Rif1^{FH}$)[12] (Supplementary Fig. 1A and B)). We then targeted the FH allele with a mini-gene encoding $Rif1^{\Delta PP1}$. As a control, following the same strategy, we also knocked-in a Rif1 wild type mini-gene ($Rif1^{TgWT}$). Thus, Cre-mediated deletion of the $Rif1^{flox}$ allele leaves either the FH-tagged $Rif1^{\Delta PP1}$, $Rif1^{TgWT}$ or the parental $Rif1^{FH}$ allele as the sole source of RIF1, effectively creating inducible FH-tagged Rif1 hemizygous cells. Upon tamoxifen-mediated Cre recombination, we have then studied the consequences of abolishing RIF1–PP1 interaction in $Rif1^{\Delta PP1/flox}$ ($Rif1^{\Delta PP1/-}$, abbreviated Rif1-ΔPP1), control $Rif1^{TgWT/flox}$ ($Rif1^{TgWT/-}$ abbreviated Rif1-TgWT) and the parental $Rif1^{FH/flox}$ ($Rif1^{FH/-}$, abbreviated Rif1-FH) cell lines. In agreement with the fact that, upon Cre induction, all the Rif1 FH-tagged alleles are hemizygous, RIF1-ΔPP1, RIF1-TgWT and RIF1-FH, are expressed at comparable levels (Fig. 1a, b and Supplementary Fig. 1C) and RIF1–PP1 interaction is undetectable in RIF1-ΔPP1 (Fig. 1c). As it was previously shown[50], we also found that expression of $Rif1^{\Delta PP1}$ leads to accumulation of hyperphosphorylated MCM4 (data not shown). Both RIF1-ΔPP1 and RIF1-TgWT have a comparable degree of chromatin-association (Fig. 1d and Supplementary Fig. 2A).

RIF1 deficiency in mESCs affects nuclear function at multiple levels. One of the features of Rif1-KO cultures is the doubling of the cell population in G2, accompanied by a decreased S-phase population (Fig. 1e)[13]. Our data show that Rif1 hemizygosity

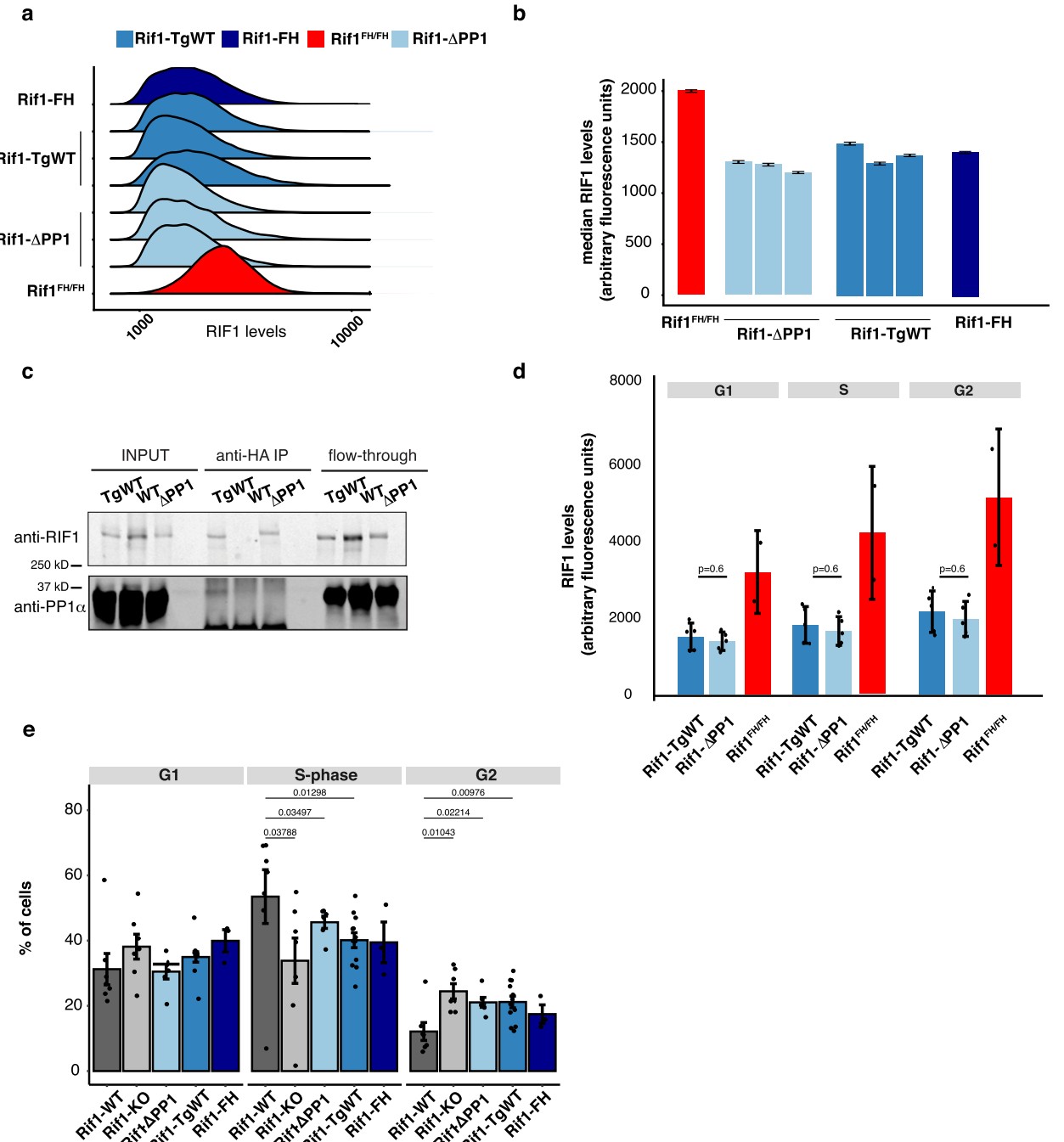

**Fig. 1 Expression levels and chromatin association of RIF1-TgWT and RIF1-ΔPP1 are comparable to those of RIF1 in hemizygous cells. a** Quantitative analysis of total levels of FH-tagged RIF1, measured by intra-cellular FACS staining. Anti-HA mouse ascites 16B12 was used to stain the indicated cell lines. *Rif1*[FH/FH]: homozygous knock-in FH-tagged RIF1, as a control of quantitative staining. The plot shows distributions of densities from HA signal, measured in arbitrary units. One representative experiment is shown. **b** Quantification from Fig. 1a. The bar plot represents the median intensities for the experiment shown and the error bars indicate 95% confidence intervals, calculated through bootstrapping with 10,000 iterations. **c** Total proteins were extracted from cells expressing untagged wild type RIF1 (WT), or from hemizygote cells expressing RIF1-TgWT or RIF1-ΔPP1 (HA tagged), and immunoprecipitated with anti-HA antibody. The input, immunoprecipitated complex and flow through were analysed by western blot with anti-mouse RIF1 affinity-purified rabbit polyclonal antibody (1240) and anti-PP1α. **d** Quantitative analysis of the levels of chromatin-associated FH-tagged RIF1 throughout the cell cycle measured by FACS staining. Cytoplasmic and nucleoplasmic proteins were pre-extracted before fixing chromatin-associated proteins. Anti-HA mouse ascites 16B12 was used to visualise FH-tagged RIF1 as in **a**. Cell cycle stage was determined by DNA quantification (DAPI staining). Means from the results from two independent experiments, each with 2 or 3 independent cell lines per genotype are summarised. The error bars indicate standard deviations. *P* values were calculated by two-sided unpaired *t* test. **e** Cell cycle distribution of the indicated cell lines, as determined by FACS quantification of EdU incorporation (S-phase) and DAPI staining (DNA amount). The mean value of three independent clones per genotype, from three experiments is shown. Error bars indicate the standard error of the mean. *P* values are calculated using two-sided Wilcoxon rank-sum test.

(*Rif1-FH* and *Rif1-TgWT*) results in an altered cell cycle similar to *Rif1* deficiency (Fig. 1e). Importantly, cell cycle distribution in both *Rif1-ΔPP1* and *Rif1-TgWT* cell lines appears comparable to *Rif1-FH* cells. These results suggest that the defective cell cycle progression of *Rif1* null cells is not attributable to altered PP1 function but to insufficient levels of RIF1.

Loss of RIF1 function also results an altered gene expression profile in mESCs[13], including the de-repression of MERVLs[38], an effect that RIF1 shares with other epigenetic and DNA replication regulators[53]. We therefore compared the level of MERVL RNA in *Rif1-WT*, *Rif1-TgWT*, *Rif1-KO* and *Rif1-ΔPP1* cells. After 4 days of deletion, MERVLs are upregulated not only in *Rif1-ΔPP1* and *Rif1-KO* cells, but, surprisingly, also in the hemizygous control (*Rif1-TgWT*, Supplementary Fig. 2B), suggesting that, as for cell cycle progression, gene expression control is also sensitive to RIF1 dosage.

**RIF1–PP1 interaction is important for the replication-timing programme.** The most conserved function of RIF1 is the control of the replication-timing programme and *Rif1-KO* cells show pronounced genome-wide changes in the temporal programme of origin firing[12,13]. As RIF1–PP1 interaction has been shown to be important, at least during the execution of the replication-timing programme in S-phase[15,48–51], expression of *Rif1ΔPP1* should affect replication timing to a similar extent to *Rif1* deletion. In agreement with this prediction, hierarchical clustering of genome-wide replication timing shows that *Rif1-ΔPP1* and *Rif1-KO* mESCs cluster together, while *Rif1+/+* (*Rif1-WT*) and control hemizygous (*Rif1-FH* and *Rif1-TgWT*) cells form a separate cluster (Fig. 2a and Supplementary Fig. 3A).

The definition of early and late replicating domains of both *Rif1-ΔPP1* and *Rif1-KO* mESCs also appears affected (the profiles are compressed around the zero, with less defined early domains —above the line—and late domains—below the line—Fig. 2b and Supplementary Fig. 3B), and a comparable fraction of the genome displays replication timing switches and changes (Supplementary Fig. 4), suggesting an analogous loss of temporal control of origin firing in both cases. Importantly, the replication timing changes induced by the expression of *Rif1ΔPP1* are not attributable to *Rif1* haploinsufficiency. In fact, the replication-timing profiles of *Rif1* hemizygous controls (*Rif1-FH* and *Rif1-TgWT*), are very similar to the wild type cells (*Rif1-WT*, Fig. 2b and c and Supplementary Fig. 4, red boxes). Despite the similarities, however, the impact of loss of RIF1 versus loss of RIF1–PP1 interaction on the replication-timing programme is quantitively not identical. *Rif1ΔPP1* expressing cells maintain a better degree of distinction between earlier and later replicating domains than *Rif-KO* (Fig. 2b, c and Supplementary Fig. 3B). These data suggest that RIF1-dependent control of the replication-timing programme could be not entirely exerted through PP1 and some other function of RIF1 could partially contribute as well.

**DNA replication timing is independent of the spatial distribution of replication foci.** DNA replication takes place in a spatially organised manner[54,55], with the distribution of replication foci correlated to the time of replication[56]. We have shown that in mouse primary embryonic fibroblasts (pMEFs), *Rif1* deficiency induces changes of both the spatial distribution of replication foci and replication timing[12]. We find a comparable effect in *Rif1*-deficient mESCs, with an increased proportion of cells displaying an early-like replication pattern (Fig. 2d) despite there being no increase in the proportion of cells in early S-phase, as judged from the analysis of DNA content (Supplementary Fig. 3C). In wild type mESCs, the early S-phase replication pattern features many small replication foci throughout the nucleoplasm (examples in Supplementary Figs. 5A, B and 6D)[57].

In Supplementary Fig. 5A, we have identified early S-phase cells by a diffuse nucleoplasmic MCM3 staining and absence of histone H3 phosphorylated on Ser10-H3S10p. In Supplementary Figs. 5B and 6D, cells in early S-phase were identified by DNA content, either by FACS sorting (Supplementary Fig. 5B, P1 and P2, empty white arrowhead), or by quantification of DAPI staining in 3D-SIM images (Supplementary Fig. 6D, early, Rif1-WT). In *Rif1-KO* cells, a diffuse distribution of replication foci (EdU or BrdU) similar to early S-phase also appears aberrantly in cells in later S-phase (Supplementary Fig. 5B, P4, empty white arrowheads and Supplementary Fig. 5C, clusters of H3S10p signal, often at chromocenters, MCM3 discrete foci, larger-mid- or smaller-late and peripheral). Normally at this stage, EdU or BrdU signal appears as discrete foci of different sizes, often associated with heterochromatin (Supplementary Fig. 5A, B, P4, full, yellow arrowheads, and Supplementary Fig. 6D, mid/late, Rif1-WT).

To study the effect of *Rif1* deficiency or expression of *Rif1ΔPP1* on the total number of replication forks and their clustering, we employed 3D-structure illumination microscopy (SIM—Supplementary Fig. 6A). Since *Rif1* deletion and expression of *Rif1ΔPP1* induce a loss of equivalence between replication foci distribution and replication timing, we could not classify early, mid and late S-phase from the EdU patterns. We therefore used DNA quantification based on DAPI staining to categorise them. *Rif1* deficiency results in an increase of the total number of replication forks throughout S-phase (Supplementary Fig. 6B) that could explain the apparent increase in the proportion of cells displaying early-like replication patterns. However, expression of *Rif1ΔPP1* does not increase the total number of replication forks per nucleus at any stage of S-phase (Supplementary Fig. 6B), yet it causes an accumulation of early-like replication patterns that is comparable to *Rif1-KO* cells (Fig. 2d). Considering the different extent of the impact of RIF1 loss (*Rif1-KO*) and loss of RIF1–PP1 interaction (*Rif1-ΔPP1*) on replication timing, this discrepancy in the effect on the total number of forks is interesting and indicates that the altered distribution of replication foci observed in both cell lines is not linked to the change of total number of replication forks. Moreover, by matching the total number of replication forks to the number of replication foci, we could not find a correlation between the number of forks per replication focus (Supplementary Fig. 6C) and the changes of distribution of replication patterns (Fig. 2d). These data indicate that the increase of the proportion of cells with early-like replication patterns observed in *Rif1-KO* and *Rif1-ΔPP1* cells is not attributable to the de-clustering of the replication forks. Finally, our analysis shows that *Rif1* hemizygosity (Rif1-hem = Rif-FH + Rif1-TgWT) has an impact on the spatial distribution of replication foci that is similar to, although milder, than *Rif1* deficiency or expression of *Rif1ΔPP1* (Fig. 2d). However, hemizygosity does not result in an increased number of total replication forks or a measurable perturbation of the replication-timing programme. These results suggest that spatial distribution of replication foci and the timing of replication can be uncoupled. In conclusion, loss or reduced RIF1 levels and loss of RIF1–PP1 interaction all impact on the distribution of replication foci, but not all affect replication timing.

**RIF1 dosage is important for nuclear compartmentalisation.** The distribution of replication foci in the nucleus reflects the spatial organisation of the underlying chromatin. As a consequence, the altered spatial configuration of replication foci in *Rif1-hem* and *Rif1-ΔPP1* cells suggests that a reduced amount of RIF1 or loss of RIF1–PP1 interaction could affect chromatin organisation similarly to what we have shown for *Rif1* null cells[13], and irrespective of their effects on replication timing. We

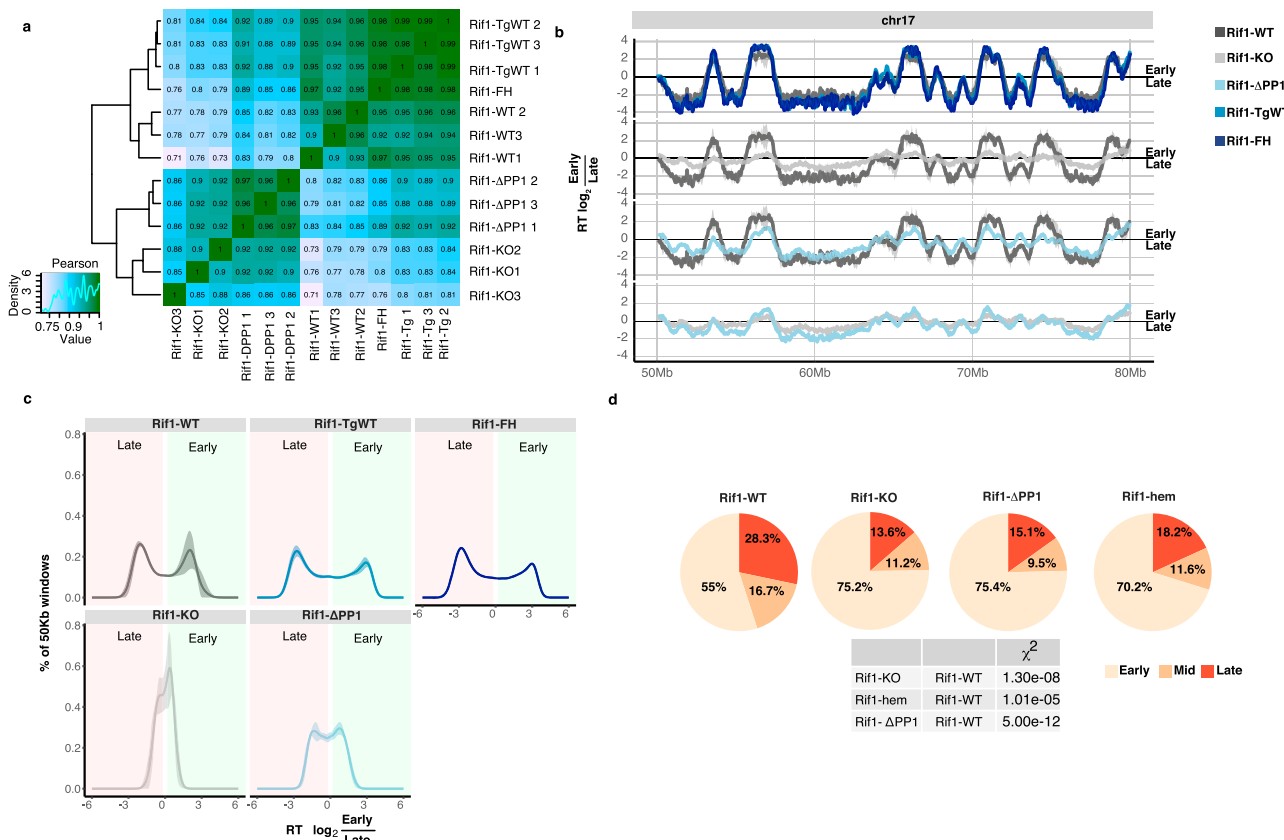

**Fig. 2 Effect of loss of RIF1–PP1 interaction on the replication-timing programme and on the spatial distribution of replication foci. a** Hierarchical cluster analysis of Pearson correlation coefficient of genome-wide replication-timing (RT) profiles between replicas, bin size 50 kb. The analysis shows preferential clustering of RT distribution from *Rif1-KO* and *Rif1-ΔPP1* lines, while RT distribution from *Rif1-WT* clusters with *Rif1-TgWT* and *Rif1-FH* lines. **b** Representative RT profile from Chromosome 17. The solid line shows the mean of three biological replicas, except for *Rif1-FH* (single, parental clone). RT scores are calculated as the $\log_2$ of the ratio between mapped reads in the early and late replicating fractions of the cell cycle over bins of 50 kb. **c** Genome-wide distribution of 50 kb genomic windows on the basis of their RT scores. Mean of three independent lines per genotype is shown, except for *Rif1-FH*. Shaded areas represent standard deviations. If the mean minus the standard deviation was <0, it was set to 0. RT scores from *Rif1-WT* and *Rif1* hemizygous lines (*Rif1-TgWT* and *Rif1-FH*) show a bimodal distribution, defining distinct early and late genomic regions. On the contrary, the distribution of RT scores from *Rif1-KO lines* shows a tendency towards a unimodal distribution, centred around zero. *Rif1-ΔPP1* lines display an increase in the windows with RT close to 0, but still a bimodal distribution of the RT values. **d** The spatial distribution of replication foci (replication patterns) was visualised by EdU and DAPI staining. Cells were pulsed for 30 min with EdU and fixed. Examples in Fig. S5A. Pie charts show the relative distribution of S-phase cells (EdU positive) between replication patterns corresponding to early, mid, and late S-phase. For each genotype, three independent lines and two separate experiments were blind-scored. As *Rif-FH* cells are a single cell line with no biological replicas (parental) and the results are very similar to the results from *Rif1-TgWT*, they were pooled (Rif1-hem). In the table, statistically significant differences are summarised. *P* values are calculated by $\chi^2$ test.

therefore analysed 3D chromatin organisation in *Rif1-KO*, *Rif1-ΔPP1* and *Rif1-hem* cells using Hi-C. We have previously shown by 4C that *Rif1* deficiency induces an increase in low-frequency contacts between TADs with different replication timing[13]. In agreement with this, our Hi-C data indicate that *Rif1* deletion increases chromatin contacts *in cis*, especially at long range (>10 Mbp) (Rif1-WT and Rif1-KO, Fig. 3a and b and Supplementary Fig. 7A), similarly and to a degree at least comparable to the depletion of the cohesin subunit SCC1 (Supplementary Fig. 7b)[58]. The contacts gained preferentially involve late-replicating genomic regions associating with early-replicating regions (Fig. 4a) and RIF1-enriched regions gaining contacts with RIF1-poor genomic regions (Fig. 4b). Loss of RIF1–PP1 interaction has the same effect as the loss of *Rif1* (Rif1-ΔPP1, Figs. 3a, b and 4a, b). Unexpectedly, chromatin architecture in *Rif1* hemizygous cells (*Rif1-FH* and *Rif1-TgWT*) shows an intermediate but reproducible degree of change. Halving *Rif1* dosage is sufficient to induce a gain of *in cis* contacts between distant genomic regions (Fig. 3a) of opposite replication timing (Fig. 4a). These changes cannot be explained by the increased fraction of cells in G2 in

*Rif1-KO*, *Rif1-ΔPP1* and *Rif1-hem* cells, as it was shown that chromosome compaction in G2/M favours the establishment of short-range interactions[59]. Consequently, the increased proportion of *Rif1-KO*, *Rif1-ΔPP1* and *Rif1-hem* cells in G2 will lead to an under-estimate of the true extent of the accumulation of long-range interactions. They suggest instead the alteration of the A/B compartmentalisation in mutant cells compared with *Rif1-WT* cells. Indeed, principle component analysis shows that *Rif1-WT* on one side and *Rif1-KO* and *Rif1-ΔPP1* on the other, display a distinctly different compartment organisation (Fig. 4c). In agreement with previous data that have reported a more "open chromatin"[12,13,17]; *Rif1* loss of function, as well as the loss of RIF1–PP1 interaction, induces an expansion of the A compartment (and corresponding contraction of the B compartment, Fig. 4d) and compartment strength is weakened (Fig. 4e and f), with increased inter-compartment interactions (Fig. 4e). Principal component analysis (PCA) of the overall A/B compartment organisation in *Rif1-TgWT* and *Rif1-FH* cells lies between with *Rif1* null and *Rif1-ΔPP1* cells on one side, and *Rif1-WT* (Fig. 4c) on the other, with an expansion of the A compartment (Fig. 4d)

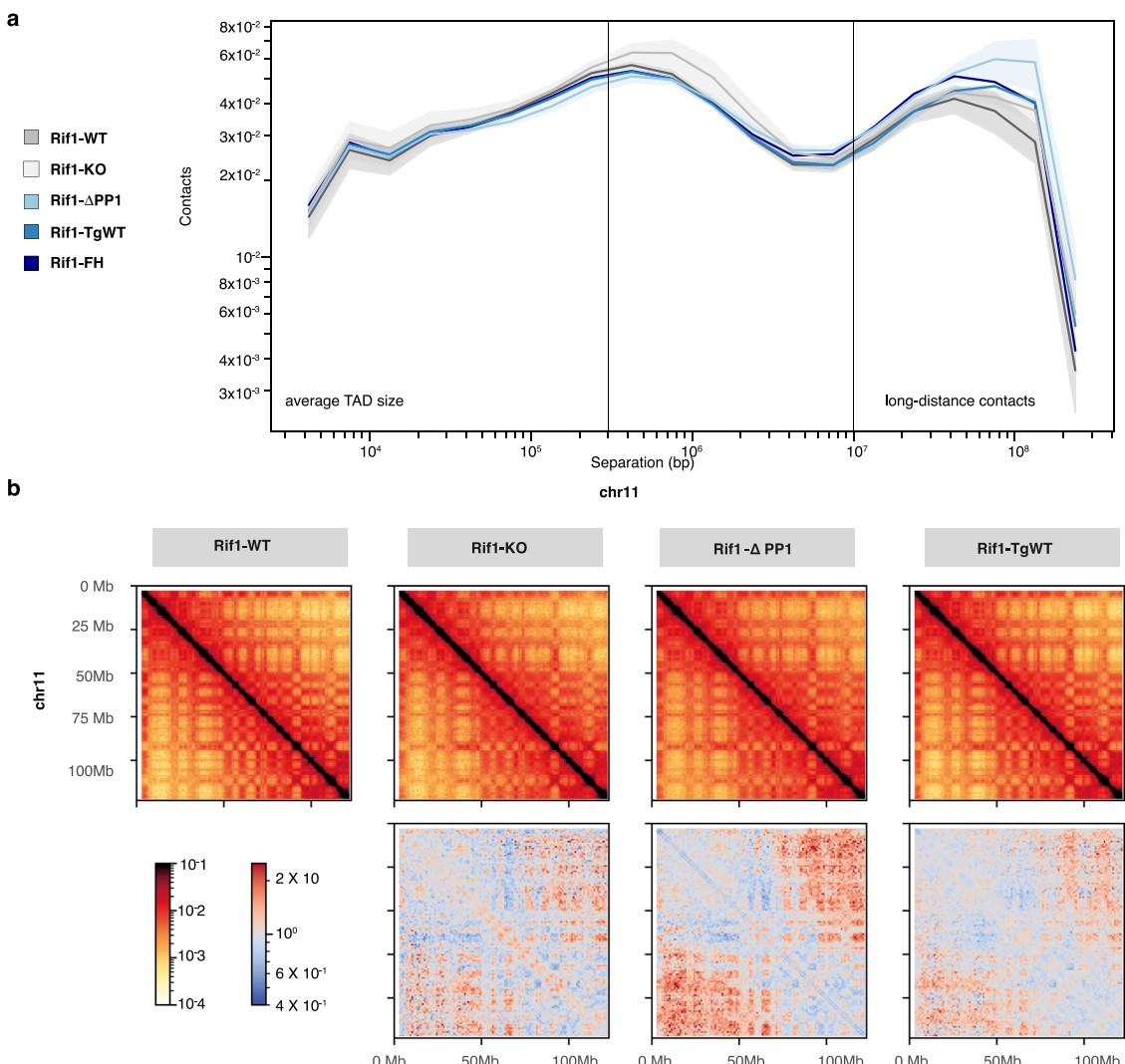

**Fig. 3 RIF1 spatially confines chromatin contacts in a dose-dependent manner. a** Normalised contact frequency versus genomic distances for Hi-C reads. Median from three biological replicas per genotype, except for *Rif1-FH*, are shown. Intra-TADs contacts (line at a median TAD's size of ~0.3 Mbp), and long-range (over 10 Mbp apart) are indicated. Shaded areas represent standard deviations. **b** Representative distribution of the median number of *in cis* chromatin contacts per indicated position (arbitrary units) within the specified region of Chromosome 11. Three independent clones per genotype were used. Upper: log(balanced HiC signals). Lower: log((balanced HiC signals (indicated line/*Rif1-WT*)). Red indicates a gain of interactions over *Rif1-WT*, while blue represents a loss.

and weakened compartmentalisation (Fig. 4e and f) that is intermediate between *Rif1-WT* and *Rif1-KO/Rif1-ΔPP1* cells. These data indicate PP1 plays a key role in RIF1-dependent control of chromatin organisation, but, also, that chromatin architecture is exquisitely sensitive to RIF1 dosage, with decreasing the levels of RIF1 inducing a progressive alteration of nuclear organisation. This is in striking contrast with the lack of any effect on the regulation of replication timing, when varying RIF1 levels.

## Discussion

The remarkable coincidence of spatial distribution and replication timing of different portions of the genome, at multiple levels of organisation and throughout evolution, has encouraged the idea of a causal relationship between nuclear architecture and replication timing. At a molecular level, their covariation—for example during cell fate determination and embryonic development—finds a confirmation in their co-dependence on RIF1. In this work, we show that both aspects of nuclear function depend upon the interaction between RIF1 and PP1. However, 3D organisation of chromatin

contacts and replication timing show a different degree of dependency on RIF1–PP1 interaction and are differentially influenced by RIF1 dosage. The loss of RIF1–PP1 interaction affects the compartmentalisation of chromatin contacts comparably to a complete loss of RIF1 function, while it partially recapitulates the effects of $Rif1^{-/-}$ on the control of replication timing. In addition, the former is sensitive to RIF1 dosage, while *Rif1* haploinsufficiency does not affect the latter. The fact that halving *Rif1* dosage affects chromatin contact organisation but not replication timing is not attributable to a lesser sensitivity of Repli-seq as compared to HiC. Replication timing measurement, even by Repli-chip, have detected differences between samples as small as 10%[60]. In summary, there is a clear division into two groups: (i) replication timing is only affected by complete loss of functional RIF1 (*Rif1-KO* and *Rif1-ΔPP1*). (ii) On the contrary, nuclear compartmentalisation, long-range chromatin contacts, replication foci spatial organisation and MERVL repression all show sensitivity to RIF1 dosage, with the effect of lack of RIF1–PP1 interaction clearly worsening the effect of hemizygosis for the long-range chromatin interactions and nuclear compartmentalisation. We hypothesise that the reason for the difference of the effect of

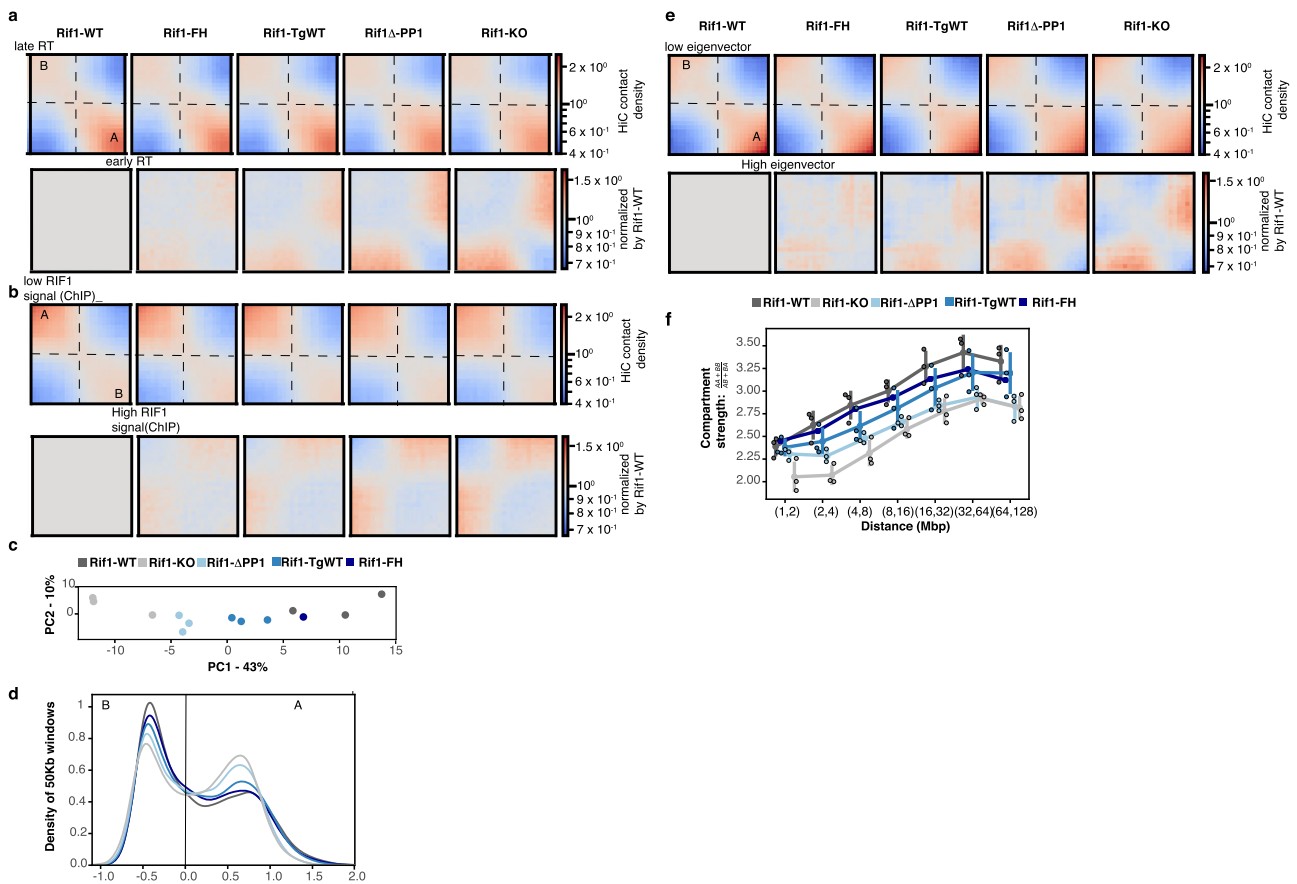

**Fig. 4 Segregation of A and B nuclear compartments is sensitive to RIF1 dosage. a** Top row: Saddle plot of Hi-C data, binned at 250 kb resolution for loci ranked by their replication timing. An increase in contacts of genomic positions of opposite replication timings is progressively more evident from *Rif1-WT* to *Rif1-KO*. The triplicates for each genotype (except for *Rif1-FH*) were combined. Lower row: Hi-C for each genotype, data normalised to *Rif1-WT*. Red indicates a gain in contacts. A and B indicates the compartments. **b** Top row: Saddle plot of Hi-C data, binned at 250 kb resolution for loci ranked by their association with RIF1[13]. An increase in contacts between RIF1-associated and RIF1-devoided genomic positions is progressively more evident from *Rif1-WT* to *Rif1-KO*. The triplicates for each genotype (except for *Rif1-FH*) were combined. Lower row: Hi-C for each genotype, data normalised to *Rif1-WT*. Red indicates a gain in contacts. A and B indicate the compartments. **c** Principle component analysis of A/B compartmentalisation for the indicated genotypes in triplicate, except for *Rif1-FH*. **d** Distribution of genomic regions of 250 kb windows between the A and B compartment. Mean of three biological replicates is shown, except for the parental line *Rif1-FH*. **e** Top row: Saddle plot of Hi-C data, binned at 250 kb resolution for loci ranked by their eigenvector values. An increase in contacts between genomic positions in different compartments is progressively more evident from *Rif1-WT* to *Rif1-KO*. The triplicates for each genotype (except for *Rif1-FH*) were combined. Lower row: Hi-C for each genotype, data normalised by *Rif1-WT*. Red indicates a gain in contacts. A and B indicates the compartments, calculated from our data. **f** Compartment strength variation with distance for the indicated genotypes. Individual values for the three biological replicates are represented by the outlined circles, except for *Rif1-FH*. The line represents the median and the bars the standard deviations.

RIF1–PP1 loss of interaction between chromatin contacts/nuclear compartmentalisation and replication foci distribution/MERVL overexpression, is that chromatin contacts/nuclear compartmentalisation are the primary features affected by loss of RIF1-dependent dephosphorylation of critical substrate/s, that directly or indirectly control them. The alteration of replication foci distribution is a more indirect way to visualise the same changes (therefore less sensitive), and modifications in gene expression (MERVL upregulation, in this case) could also be an indirect consequence of these architectural changes, as we had already hypothesised[12,13]. We have indeed shown that the transcriptome is only altered after a few cell cycles in the absence of RIF1, while nuclear architecture changes are an immediate consequence of RIF1 absence, in the first cell cycle after *Rif1* deletion[13]. Analysis of the effects of the loss of RIF1–PP1 interaction is extremely complex. Chronic deletion of *Rif1* causes cell cycle arrest[12], cell death[13] and genome instability[19]. If and when a *Rif1*−/− cell lines can be obtained, it is through selection of survivors that are transcriptionally and genomically unstable and difficult to control for. Since *Rif1-KO* and *Rif1-ΔPP1* share most of the cellular phenotypes,

homozygous knock-in of *Rif1-ΔPP1* would have the same issues as *Rif1*−/− cells. On the other hand, *Rif1* over-expression can only be analysed in transient, as it is toxic. Therefore, although *Rif1-hem* present the inconvenience of a basal level of deregulation of most of RIF1 functions, this system represents the best-controlled situation to address the key question of the role of RIF1–PP1 interaction.

RIF1 is known to multimerise[45,47,61] and to interact with the nuclear lamina[13,42]. RIF1 multimers could act as a sub-stochiometric platform, interacting with different regulators of replication timing, in addition to PP1. In this case, the consequences of the complete loss of RIF1 function on the replication-timing programme would amount to the sum of perturbation of multiple pathways that control the timing of origin activation. For example, RIF1-ΔPP1 may only specifically interfere with the PP1-dependent control of DDK (Dbf4-dependent kinases) activity at origins, while other RIF1 interactors may contribute to the epigenetic control of origin activation. Proteins associated with RIF1 are enriched for chromatin and epigenetic regulators[52], and the contribution of histone modifiers to the control of replication timing has long been

recognised[60,62–66]. However, an understanding of the effect of *Rif1* deletion on the epigenetic landscape is still missing, leaving this hypothesis currently hard to test[12,13,38,67]. In the context of chromatin architecture, RIF1 multimers could directly participate in the creation of local scaffolds that restrict chromatin mobility or could regulate other proteins with this role. In either case, a reduction of RIF1 dosage could have structural, quantitative consequences.

Our results identify RIF1 as a molecular link, a point of convergence and co-regulation. We propose that RIF1, specifically, and not generic nuclear architecture, coordinates the replication-timing programme with nuclear 3D organisation. In agreement with this view, recent data show that cohesin and CTCF are not involved in the regulation of replication timing[6,68] and that the definition of A/B compartments and Early/Late replicating domains is uncoupled at the time of zygotic genome activation in zebrafish[69] and during the first cell cycles of human ESCs differentiation[70]. Altogether, these data suggest that replication timing and nuclear architecture, or at least 3D organisation of chromatin contacts and spatial distribution of replication foci, are not linked by a causative relationship. Yet, they are coregulated, both during cell cycle and embryonic development, and RIF1 is a point of convergence. Having established this is an important step to start addressing the fundamental question of why this coordination is important. During embryonic development in different organisms, for example in *Drosophila melanogaster*, replication timing[16] and TADs definition both emerge around the time when zygotic transcription starts[71]. Could uncoupling these two events have consequences on gene expression? We can alter chromatin organisation, leaving replication timing intact, by halving RIF1 dosage. This affects cell cycle progression and the repression of MERVLs (this work). In a complementary approach, it has been shown that alteration of replication timing by overexpression of limiting replication factors during early *Xenopus laevis* development, that presumably leaves nuclear architecture intact, affects the onset of zygotic transcription and the transition into gastrulation[72]. It is therefore tempting to speculate that the covariation of replication timing and nuclear architecture could be important to coordinate gene expression and the choice of origins of replication.

## Methods

**Mouse ESC derivation**. Mouse ESC cells were derived as described in ref. [13], with the addition of 1 μM MEK inhibitor PD0325901 and 3 μM GSK3 inhibitor CHIR99021 (MRC PPU Reagents and Services, School of Life Sciences, The University of Dundee) in the culture media, from the start of the protocol.

Rif1$^{FH/flox}$ Rosa26$^{Cre-ERT/+}$ mESCs were derived by crossing Rif1$^{FH/flox}$ Rosa26$^{Cre-ERT/Cre-ERT19}$ with Rif1$^{FH/FH12}$ mice. The Rif1$^{FH}$ allele was specifically targeted in the parental line Rif1$^{FH/flox}$ Rosa26$^{Cre-ERT/+}$ (Rif1-FH). Integrants were selected by hygromycin resistance. The targeting vector encodes a codon-optimised cDNA of RIF1 (exon 8–exon 36). Hygromycin-resistant colonies were screened for correct targeting of the Rif1$^{FH}$ allele by Southern blot (EcoRV digest) and using a PCR-amplified probe (primers in Supplementary Information table "Primers").

**Cell manipulation**. mESCs were grown at 37 °C in 7.5% CO$_2$ in Knockout DMEM (Gibco 10829-018), containing 12.5% heat-inactivated foetal bovine serum (Pan-Biotech), 1% non-essential amino acids (Gibco 11140-035), 1% penicillin/streptomycin (Gibco 15070063), 0.1 mM 2-Mercaptoethanol (Gibco 31350-010), 1% L-glutamine (Gibco 25030024)), supplemented with 1 μM PD0325901 and 3 μM CHIR99021 and 20 ng/ml leukaemia inhibitory factor (LIF, EMBL Protein Expression and Purification core facility).

Experiments were carried out each time from a frozen vial of cells, at least two passages after thawing. 5.2 × 10$^6$ cells for Rif1-WT and 6.5 × 10$^6$ for Rif1-KO lines, per 15 cm plate (or the equivalent for different sized plates) were plated at day zero, when treatment with 200 nM 4-hydroxytamoxifen (OHT, Sigma H7904) started. Fresh medium with OHT was added after 48 h. Cells were collected about 96 h after starting OHT treatment.

**Replication timing analysis**. Cells were pulsed for 2 hours with 100 μM BrdU, collected and fixed in 70% ethanol. Processing was as described in ref. [73]. Fastq files were aligned using Bowtie2 version 2.2.6 on mm10 as a reference genome. SAM files were converted into BAM files and sorted using Samtools version: 1.3.1. bamCompare version 3.1.3 was used to create bedgraph files with 50 and 1 kb

binning of the log$_2$ ratio of the early and late fraction. Duplicated reads were excluded from the computation of the bedgraph files as well as reads mapped on XY chromosomes. The two fractions were normalised as reads per millions (RPM). Plots and data manipulation were carried out using R version 3.5.1. The original names of the cell lines used in these experiments, included in the name of the Repli-seq raw files are: RFHF14 = Rif1-FH, 14 tgWT A7 = Rif1-TgWT 1, 14 tgWT H4 = Rif1-TgWT 2, 14 tgwt H6 = Rif1-TgWT 3, 14 ΔP G11 = Rif1-ΔPP1 1, 14 ΔP H1 = Rif1-ΔPP1 2, 14 ΔP H2 = Rif1-ΔPP1 3, mESC B = Rif1-WT 1, mESC F = Rif1-WT 2, mESC H = Rif1-WT 3, mESC 5 = Rif1-KO 1, mESC 18 = Rif1-KO 2, and mESC 24 = Rif1-KO 3.

**Intra-cellular FACS staining for $^{HA}$RIF1**. After 4 days of OHT treatment, cells were collected and counted. 3 × 10$^6$ cells were fixed in 400 μl of DPBS/2% paraformaldehyde (Sigma P-6148) for 10 minutes at room temperature shaking. Paraformaldehyde was then diluted to 0.2% and next cells were washed in cold DPBS. After 2 minutes permeabilisation in 200 μl PBS-Triton X-100 0.1%, cells were incubated 5 minutes in saponin solution (COMPONENT E from kit C10424, Thermo Fisher Scientific) at room temperature and anti-HA antibody (Covance monoclonal HA.11 clone 16B12 #MMS-101R, RRID:AB_291262) was added at 1:500. After 1 hour at room temperature rotating, cells were washed twice in DPBS/2% FBS, resuspended in 200 μl of saponin solution with goat anti-mouse Alexa Fluor 647 1:1000 (Thermo Fisher Scientific A-21235, RRID:AB_2535804) and incubated for 1 hour rotating in the dark. After washing twice samples were resuspended in 400 μl of saponin solution with DAPI 2.5 g/ml (Thermo Fisher Scientific D1306) and analysed on an LSR II FACS (BD). .Data were processed using R version 3.5.1. The confidence intervals (CI) of the median shown in Fig. 1b were calculated by bootstrap.

For the FACS analysis of RIF1's chromatin association, the samples were processed as above, except, fixation was preceded by 3 minutes incubation in CSK buffer (25 mM HEPES pH 7.4, 50 mM NaCl, 1 mM EDTA, 3 mM MgCl$_2$, 300 mM sucrose, 0.5% Triton X-100 and complete protease inhibitor cocktail tablet). Pre-extracted cells were subsequently fixed in 3% PFA/sucrose for 30 minutes at room temperature shaking. Example of the gating strategy in Fig. S8.

**Cell cycle distribution analysis**. After 4 days of OHT treatment, cells were pulsed for 30 minutes with 10 μM EdU (Invitrogen A10044). Cells were then washed with cold DPBS (Thermo Fisher Scientific 14190094), collected, counted and fixed in 75%. EtOH Samples were kept at −20 °C for at least overnight. 7.5 × 10$^5$ cells were then processed for click-chemistry detection of EdU. After washing in cold DPBS, cells were permeabilised in DPBS/1% FBS/0.01% Triton X-100 (Sigma 93426-250ML) for 30 minutes on ice. After washing twice, cells were incubated in 900 μl of DPBS with 10 mM Na-Ascorbate (Sigma A7631-25G), 1 μM Alexa Fluor 647 Azide (Thermo Fisher Scientific A10277) and CuSO$_4$ 0.1 M (Sigma C1297) for 30 minutes at room temperature in the dark, rotating. Cells were washed in DPBS/1%FBS/0.5% Tween 20 (Sigma P9416-100ML) for 10 minutes and then twice in cold DPBS/1% FBS. After 1 hour incubation in 300 μl of DPBS/1%FBS/DAPI 2.5 g/ml (Thermo Fisher Scientific D1306), the samples were analysed using an LSR II FACS (BD). The data acquired were analysed using Flowjo software and plotted in R 3.5.1. To calculate the percentages of cells in early, mid and late S-phase in Supplementary Fig. 2C, we have defined the S-phase substages based on the intensities of the PI/EdU signals in the wild type, drawn the gates and applied them to all the samples[12]. Gating strategy in Figs. S9 and S10.

**Reporting summary**. Further information on research design is available in the Nature Research Reporting Summary linked to this article.

## Data availability
The HiC data have been deposited and are available at accession GEO: GSE148244. The replication timing data have been deposited and are available at NCBI: https://www.ncbi.nlm.nih.gov/sra/?term=PRJNA545793. Source data are provided with this paper.

## Code availability
The codes employed in the analyses of the data associated to this manuscript are available through the Buonomo lab github: https://git.ecdf.ed.ac.uk/buonomo/gnan-et-al.-2021.

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

## Acknowledgements
We would like to acknowledge Martin Waterfall from the IIIR Flow Cytometry Core Facility, University of Edinburgh; David Kelly from the COIL facility, WTCCB, University of Edinburgh; Vladimir Benes and the Genomic Core Facility at EMBL Heidelberg; Philip Hublitz from Gene expression Facility, EMBL Monterotondo; Violetta Parimbeni for mouse husbandry, EMBL Monterotondo. We thank Chunlong Chen, Institut Curie, for critically reading the manuscript. S.G. was funded by ERC consolidator award 726130 to S.C.B.B.; E.C. was supported by the Erasmus Programme. E.E. received funding from the European Union's Horizon 2020 research and the Marie Skłodowska-Curie Individual Fellowship grant agreement No. 660985 and from the ERC consolidator award 726130 to S.C.B.B. I.M.F. was funded by the Darwin Trust of Edinburgh. S.C.B.B. thanks the ERC, D.M.G. thanks NIH grant GM083337, W.A.B. is funded by a Medical Research Council University Unit programme grant [MC_UU_00007/2]. M.C.C. was funded by the Deutsche Forschungsgemeinschaft (DFG, German Research Foundation) —Project-ID 393547839—SFB 1361 TP06 and DFG grant CA 198/9-2.

## Author contributions
S.G. has created the cellular system, performed the majority of the experiments and the bioinformatics analysis. I.M.F. has helped with HiC experiments and analysis. K.N.K. has performed the replication timing measures. E.C. and N.C. have stained and scored with SG replication-timing patterns. P.W., A.R. and A.M. have stained cells, acquired 3D-SIM images and performed the analysis. E.E. has analysed MERVL expression. W.A.B., D.M.G., M.C.C. have supported the work with personnel, resources, scientific discussions and critical reading of the manuscript. S.C.B.B. has conceived the project, performed some of the experiments and written the manuscript.

## Competing interests
The authors declare no competing interests.
