## [Peer Review File · Nature Communications]

REVIEWER COMMENTS

Reviewer #1 (Remarks to the Author):

The work described by Gnan et al questions the longstanding observation that replication timing (RT) control and the spatial organization of the genome into different compartments seem to be co-regulated based on genome-wide analyses. However, the direct causal link has never been proven. One potential major factor involved in this co-regulation could be RIF1, a complex protein already known to be involved in a large variety of functions including DNA repair, telomere length regulation in yeast, cytokinesis, epigenetic and finally the control of replication timing. This last function is known to be regulated through its interaction with PP1, a phosphatase that counteracts the activity of CDKs and DDKs involved in origin firing.

In this paper, the authors use an elegant genetic approach to decipher the specific role of PP1 interaction with Rif1 in both replication timing and nuclear organization. Their results demonstrate that both RT and nuclear organization depend upon RIF1-PP1 interaction. This result is new for nuclear organization. Regarding its role in RT, this study also shows for the first time that loss of RIF1-PP1 interaction does not fully recapitulate the effects of the deletion of Rif1 suggesting a complementary role of Rif1 in the control of replication timing, probably at the level of "epigenetic organization".

Another interesting observation is that by contrast to RT, nuclear organization is highly sensitive to RIF1's dosage. One key and very new observation is that Rif1 hemizyosity has an impact on the spatial distribution of replication foci but does not affect the RT program. This last result suggests that nuclear organization of replication foci in S-phase, is not a key regulatory factor of the normal and sequential activation of replication origins. This last result puts forward the robustness of the RT control and suggests that many layers of regulation do exist.

Altogether, this paper reveals a complex relationship between nuclear organization and the control of replication timing. It is important for the community to have a clear vision of how Rif1 and nuclear organization contribute to RT control. This paper provides a much better understanding than a simple vision of a direct link between the two nuclear properties. Despite careful reading, I could not find any major criticism to be addressed. The experiments are very well-done and controlled. All the conclusions drawn by the study are therefore well-established. For all these reasons, I think that the paper is an important study that deserves to be published in Nature communications. I therefore strongly support its publication.

Reviewer #2 (Remarks to the Author):

Rif1 is a multifunctional protein with roles in replication timing, telomeres and DNA repair. The authors use state of the art techniques to shed new insight into Rif1 function, replication timing and 3D genome organization. Main (novel) conclusions of the paper are (i) Rif1 can maintain some replication timing in the absence of its PP1 interaction domain, thus Rif1 itself must be able to partially regulate replication timing independently of PP1. (ii) Rif1 regulates 3D genome organization in a dosage dependent manner and at least in part through its interaction with PP1.

The authors present a set of well planned and executed experiments using solid methodology and excellent analysis pipelines. While I find some of the data could be interpreted differently, the article certainly merits consideration at Nature Comm. Please find my comments on the article below.

Major points

1. Figure 2/ replication timing

RIF1 recruits PP1 to replication sites which in turn dephosphorylates MCM complex at sites that are phosphorylated by Cdc7 (Pubmed: 24532715). What is the level of MCM complex phosphorylation (at these Cdc7 sites) in your cell types (delta PP1 vs WT/FH)? In other words, in the [presumed]

absence of Rif1-PP1 interaction, is the MCM complex phosphorylation similar to a RIF1 KO level?

Fig2 A,B and C looks like RIFdeltaPP1 has a replication timing profile that is an intermediate between an WT and a KO (closer to the knockout). While the authors conclusion below is probable,

“RIF1-dependent control of the replication-timing program could be not entirely exerted through PP1 and some other function of RIF1 partially could contribute as well.”

an alternative explanation could be that PP1 can get to replication sites independently of RIF1 or (more likely) that the residual interaction between RIF1 and PP1 is sufficient to produce an intermediate phenotype. Can the authors unequivocally refute residual *in vivo* interactions?

2. Figure 3: Hi-C

The three replicates of the Rif1-WT display variation (Common in Hi-C assays). Figure 3B cannot be interpreted accurately without being able to see this variation (and how it compares to the differences observed in the examined/experimental cell types).

Can the authors produce a contact frequency heatmap (exactly as in Figure 3B lower panel in blue and red) for Rif-WT replicate 1 against replicate 2 and 3 (or 2 vs 1 and 3 etc..)?

The heatmaps in figure 3B are extremely similar to each other.

Are the differences observed in Figure 3A and B statistically significant?

More importantly, are they biologically significant? Can the authors please comment?

3. Figure S5 could be moved to the main figures, since it supports a main conclusion. The conclusion drawn from this figure is not clear to me: the authors state that “However, expression of Rif1DPP1 does not alter the total number of replication forks per nucleus” but FigS5B shows that each of the box plots are statistically different from each other. Perhaps they mean that despite a statistical difference, it is not biologically significant?

In Fig S5B, the hemizygote (Rif1-TgWT) does NOT have an intermediate phenotype between the WT and KO, which is surprising. Is this a problem of having just two replicates? Would the conclusion differ if more replicates were added?

In previous publications where this method is used, how many cells and foci were scored? Do the authors have similar statistical power to draw their conclusion?

4. While I understand the difficulty of generating a Rif1-deltaPP1 model without hemizygosity, the results are somehow clouded by the use of a hemizygote and this needs to be addressed in the discussion.

5. Finally, one of the authors conclusion hinges on the fact that hemizygosity of Rif1 (or dosage) impacts 3D contacts observed by Hi-C but does not impact replication timing. While this is very attractive and supported by their data, it is important to note that Hi-C and replication timing experiments are two different assays with different sensitivities. The authors need to verify that replication timing assay can in fact pick up on intermediate phenotypes with a similar sensitivity as the Hi-C.

Minor points

1. Could the authors specify if there are any known defects to the deletion of the PP1 interaction sites of Rif1? Can this deletion potentially impact other interactions of Rif1? (It seems it doesn't impact DNA binding but how about multimerization?). Please discuss briefly in the article.

2. Figure 1F: Could the authors please represent this figure as absolute amounts of MERVL RNA normalized to the house keeping gene and include the individual data points? The retrovirus DNA

is usually repressed in cells, thus very little MERVL RNA would be present in WT cells; 20 fold more of almost nothing, is, still almost nothing. If this is the case, it would not be appropriate to say there is significant gene expression changes or that MERVL is de-repressed.

3. Please clarify and explain this sentence: " The replication profiles of both Rif1-DPP1 and Rif1-KO mESCs appear similarly compressed around the zero (Fig. 2B and S2B)...".

This is one example where the writing could be improved to appeal to a wider audience. I think the authors findings have implications beyond the field of DNA replication and thus the article will benefit from appealing to a wider audience who are not familiar with specific assays.

4. Figure 4: Perhaps best to label Figures A, B, C, D and E in order.

5. What is the y-axis scale in Fig S5B? Is it log2?
Can the authors present S5B and C in linear scale?

Reviewer #3 (Remarks to the Author):

Rif1 is a large polypeptide that has been linked to the control of DNA replication timing, double strand break repair and chromatin organization in different experimental systems. Previous studies have shown that at least part of Rif1 functions require its interaction with protein phosphatase 1 (PP1), which counteracts the phosphorylation of MCM proteins at late replication origins. This new manuscript from the Buonomo lab reports a set of experiments to address how chromatin organization and replication timing (RT) are affected by changes in Rif1 genetic dosage or by the loss of the interaction between Rif1 and PP1.

The authors have genetically modified mouse embryonic stem cell lines to knock-in either WT Rif1 or a mutant version designed to disrupt its interaction with PP1. This is done in a cellular system in which the second Rif1 allele is conditionally inactivated by Cre-Lox (Figures 1 and S1). They show that chromosomal RT profiles are affected in Rif1- Δ PP1 cells almost as much as in Rif1 KO cells, but not by having Rif1 in hemizyosity (Figures 2 and S2-S3). The spatial distribution of replication foci is inferred by the analyses of patterns of EdU foci and other markers (Fig S4), as well as the number and clustering of replication forks by 3D-structure illumination microscopy (Fig. S5). Finally, chromatin organization Hi-C assays (Figures 3-4) reveal that the loss of Rif1-PP1 interaction alters chromatin contacts as much as Rif1 KO cells. From this combination of approaches, the authors propose that both nuclear architecture and RT depend on the Rif1-PP1 interaction, and that these two aspects can be partially uncoupled.

The lack of a strict correlation between 3D chromatin contacts and RT is interesting and in line with the recent findings that alterations in cohesin or CTCF have little or no impact on RT. In my opinion, this study provides a moderate advance in our understanding of the co-regulation and causal links between chromatin organization, RT, and transcriptional programs. The bigger mechanistic question at this stage would be the molecular target(s) of Rif1-PP1 that regulate chromatin contacts and nuclear compartmentalization.

General comments:

1. Some of the Results are a bit difficult to follow. A possible reason is that the study is apparently focused on the Rif1- Δ PP1 mutant version, which mimics many of the phenotypes caused the complete ablation of Rif1 and confirms the importance of Rif1-PP1 interaction in mESCs. But the conclusion that 3D chromatin contacts and spatial organization of DNA replication factories can be separated from RT is largely derived from the analyses of WT Rif1 in hemizyosity (sufficient to change chromatin conformation without affecting RT). My impression is that the information does not always flow naturally, and this may be accentuated in some cases by the figure design (see below).

2. The experiments showing alterations in the patterns of replication foci (Figure S4; quantification

pie plots in Figure 2D) are inherently complicated because changes in Rif1 levels alter the markers normally used to call “early S” or “late S” patterns. A related limitation occurs in the analyses of fork activity by SIM, as acknowledged in the text (Fig S5; “Since Rif1 deletion and expression of Rif1-DPP1 induce a loss of equivalence between replication foci distribution and replication timing, we could not analyse early, mid and late S phase separately”). In both cases, the results would be more conclusive if the IF (Figure S4) and SIM experiments (Fig S5) were performed in cells synchronized and collected at different time points along S phase. A simple possibility would be to sort cells by FACS based on their DNA content: regardless of the duration of S phase in each case, three cell fractions sorted with increasing amounts of DNA (between 1C and 2C) would necessarily correspond to early, mid and late S phase.

3. The Hi-C results should be presented and discussed with more clarity for the non-specialist. For instance, the results with Rif1- Δ PP1 (Figure 3A-B) are not mentioned initially (page 8, last paragraph) but are discussed shortly afterwards, creating a couple of confusing sentences. Also, in the saddle plots in Figure 4A, loci are ranked by their replication timing. Therefore, compartmentalization is not directly derived from this analysis (Figure legend reads “A and B indicates the compartments”), but rather inferred from previous studies, correct?

Specific suggestions about the Figures:

4. In Figure 1A-B, the expression levels of HA-Rif1 in the different cell lines are shown in a rather unusual manner (distribution of intracellular flow cytometry staining with anti-HA, in arbitrary units). What would be the background signal in this assay, e.g. without secondary antibody? Why not include also a simple immunoblot for the different clones (as in the example shown in Figure 1C?)

5. Figure 1D and Supp 1D contain histograms showing two replicates of the same experiment. I would recommend: (a) showing the flow cytometry plots for a representative experiment; in this case, a control with another protein whose chromatin association fluctuates in the cell cycle (e.g. one of the MCM proteins) could be shown to validate the technical approach; (b) showing the histograms with the quantification and statistics of three replicates.

6. Figure 1F is not very informative (esp. given the large variation between replicates) and a bit separate from the rest of the study. It could be shown as Supplementary material.

7. Figure 2B could benefit from a change in colors, particularly for Rif1-WT and Rif1-FH lines (very similar in the current version).

8. In current Figure S4 or its future equivalent, showing examples of Rif1-TgWT and Rif1- Δ PP1 nuclei would improve the message.

We would like to thank the reviewers for their very constructive and insightful comments. Please find below a point-by-point response.

Reviewer #1:

The work described by Gnan et al questions the longstanding observation that replication timing (RT) control and the spatial organization of the genome into different compartments seem to be co-regulated based on genome-wide analyses. However, the direct causal link has never been proven. One potential major factor involved in this co-regulation could be RIF1, a complex protein already known to be involved in a large variety of functions including DNA repair, telomere length regulation in yeast, cytokinesis, epigenetic and finally the control of replication timing. This last function is known to be regulated through its interaction with PP1, a phosphatase that counteracts the activity of CDKs and DDKs involved in origin firing.

In this paper, the authors use an elegant genetic approach to decipher the specific role of PP1 interaction with Rif1 in both replication timing and nuclear organization. Their results demonstrate that both RT and nuclear organization depend upon RIF1-PP1 interaction. This result is new for nuclear organization. Regarding its role in RT, this study also shows for the first time that loss of RIF1-PP1 interaction does not fully recapitulate the effects of the deletion of Rif1 suggesting a complementary role of Rif1 in the control of replication timing, probably at the level of "epigenetic organization".

Another interesting observation is that by contrast to RT, nuclear organization is highly sensitive to RIF1's dosage. One key and very new observation is that Rif1 hemizyosity has an impact on the spatial distribution of replication foci but does not affect the RT program. This last result suggests that nuclear organization of replication foci in S-phase, is not a key regulatory factor of the normal and sequential activation of replication origins. This last result puts forward the robustness of the RT control and suggests that many layers of regulation do exist.

Altogether, this paper reveals a complex relationship between nuclear organization and the control of replication timing. It is important for the community to have a clear vision of how Rif1 and nuclear organization contribute to RT control. This paper provides a much better understanding than a simple vision of a direct link between the two nuclear properties. Despite careful reading, I could not find any major criticism to be addressed. The experiments are very well-done and controlled. All the conclusions drawn by the study are therefore well-established. For all these reasons, I think that the paper is an important study that deserves to be published in Nature communications. I therefore strongly support its publication.

We would like to thank this reviewer for the support.

Reviewer #2 (Remarks to the Author):

Rif1 is a multifunctional protein with roles in replication timing, telomeres and DNA repair. The authors use state of the art techniques to shed new insight into Rif1 function, replication timing and 3D genome organization. Main (novel) conclusions of the paper are (i) Rif1 can maintain some replication timing in the absence of its PP1 interaction domain, thus Rif1 itself must be able to partially regulate replication timing independently of PP1. (ii) Rif1 regulates 3D genome organization in a dosage dependent manner and at least in part through its interaction with PP1.

The authors present a set of well planned and executed experiments using solid methodology and excellent analysis pipelines. While I find some of the data could be interpreted differently, the article certainly merits consideration at Nature Comm. Please find my comments on the article below.

Major points

1. Figure 2/ replication timing

RIF1 recruits PP1 to replication sites which in turn dephosphorylates MCM complex at sites that are phosphorylated by Cdc7 (Pubmed: 24532715). What is the level of MCM complex phosphorylation (at these Cdc7 sites) in your cell types (delta PP1 vs WT/FH)? In other words, in the [presumed] absence of Rif1-PP1 interaction, is the MCM complex phosphorylation similar to a RIF1 KO level?

We could not answer this question by simply immunoblotting for phosphorylated MCM4 because we could not find an antibody that reliably showed either MCM4 shift or specificity for phospho-MCM4 in mouse cells (the majority of the antibodies work well for human phospho-MCM4). However, we have unpublished data that we are willing to share with the reviewer, should he/she desire so, that form the basis of new projects ongoing in the lab. We have performed a phospho-proteome analysis in collaboration with the groups of Prof. Blow, Dundee University, Dr. Ly, University of Edinburgh, and Prof. Donaldson, University of Aberdeen. We have compared Rif1-WT, Rif1-KO and

Rif1-ΔPP1 cells and found that in both Rif1-KO and Rif1-ΔPP1 cells, MCM4 is specifically found hyperphosphorylated. As these data have been acquired in the context of a screen and further verification have been done on other candidates, we would prefer not to show these data in the manuscript. However, we have added a sentence to refer to these as “data not shown”.

Fig2 A,B and C looks like RIFdeltaPP1 has a replication timing profile that is an intermediate between an WT and a KO (closer to the knockout). While the authors conclusion below is probable,

“RIF1-dependent control of the replication-timing program could be not entirely exerted through PP1 and some other function of RIF1 partially could contribute as well.”

an alternative explanation could be that PP1 can get to replication sites independently of RIF1 or (more likely) that the residual interaction between RIF1 and PP1 is sufficient to produce an intermediate phenotype. Can the authors unequivocally refute residual in vivo interactions?

The IP shown in Figure 1C indicates that the interaction is abolished. In addition, the unpublished data mentioned above support the idea that the mutations are sufficient to abolish the interaction between PP1 and RIF1, as hyperphosphorylated proteins identified in the Rif1-KO line were also found in Rif1-ΔPP1 lines.

2. Figure 3: Hi-C

The three replicates of the Rif1-WT display variation (Common in Hi-C assays). Figure 3B cannot be interpreted accurately without being able to see this variation (and how it compares to the differences observed in the examined/experimental cell types).

Can the authors produce a contact frequency heatmap (exactly as in Figure 3B lower panel in blue and red) for Rif-WT replicate 1 against replicate 2 and 3 (or 2 vs 1 and 3 etc..)?

We have added the requested data in Fig. S7A

The heatmaps in figure 3B are extremely similar to each other.
Are the differences observed in Figure 3A and B statistically significant?
More importantly, are they biologically significant? Can the authors please comment?

In order to illustrate their biological significance and put our data in context, we have compared the magnitude of the changes of long-range chromatin interactions caused by Rif1-KO and Rif1-ΔPP1 with those induced by acute depletion of Scc1 or Ring1B. To this end, we have re-analysed the data from Rhodes et al. 2020, using our pipeline (Fig. S7B). As the effect of chromatin organisation changes caused by Rif1 deletion or by the loss of RIF1-PP1 interaction is comparable, if not greater, than the alterations induced by the loss of these well-studied chromatin organisers, this direct comparison highlights the importance of our data.

3. Figure S5 could be moved to the main figures, since it supports a main conclusion. The conclusion drawn from this figure is not clear to me: the authors state that “However, expression of Rif1DPP1 does not alter the total number of replication forks per nucleus” but FigS5B shows that each of the box plots are statistically different from each other. Perhaps they mean that despite a statistical difference, it is not biologically significant?

The data is now in Fig. S6B. What we meant to say is that in Rif1-KO cells there seem to be more forks than in all the other genotypes, accompanied by the changes in RT, chromatin contact and compartmentalisation. However, in Rif1-ΔPP1 and in Rif1-hem, there are less replication forks, yet, the Rif1-ΔPP1 have changes in RT and chromatin organisation, while the Rif1-hem only have changes in chromatin organisation. In summary, there is no correlation between the number of forks and the changes in RT and in chromatin organisation. If the differences have a biological significance is hard to tell, we did not comment. We have modified the text to reflect more accurately what is shown in the figure.

In Fig S5B, the hemizygote (Rif1-TgWT) does NOT have an intermediate phenotype between the WT and KO, which is surprising. Is this a problem of having just two replicates? Would the conclusion differ if more replicates were added?

A similar effect can be seen in Fig. 2D, where the percentage of cells displaying aberrant patterns is very similar in Rif1-KO, Rif1-TgWT and Rif1-ΔPP1.

In the discussion, we propose that the fact that HiC of Rif1-hem shows an intermediate level of perturbation, while the distribution of replication foci (Fig. S5) or replication forks (Fig. S6) of Rif1-hem is very close to Rif1-KO, is due to a difference in the sensitivity of these two approaches. If RIF1 levels primarily affect, directly, the organisation of chromatin contacts or the distribution of chromatin in subnuclear compartments, the detection of the distribution of replication foci or forks is an indirect way of visualising chromatin contacts. Of the two

approaches, the HiC would therefore be more sensitive as it adds up contacts from a large population of cells in comparison to a single cell-based analysis.

In previous publications where this method is used, how many cells and foci were scored? Do the authors have similar statistical power to draw their conclusion?

In the current study, we have analysed a large number of cells (in total 414 cells from independent replicates, with approximately 400,000 foci). Therefore, we do not expect to obtain different results by increasing the number of cells. We have used roughly 5 to 8 times more cells per condition than it was done in previous studies (Natale et al, 2017, Nat Comm 8,). A sentence was added in the methods to clarify this.

4. While I understand the difficulty of generating a Rif1-deltaPP1 model without hemizygoty, the results are somehow clouded by the use of a hemizygote and this needs to be addressed in the discussion.

A paragraph has been added in the discussion to consider this point. I would also add that the additional information obtained from the hemizygotes is also an asset to this study.

5. Finally, one of the authors conclusion hinges on the fact that hemizygoty of Rif1 (or dosage) impacts 3D contacts observed by Hi-C but does not impact replication timing. While this is very attractive and supported by their data, it is important to note that Hi-C and replication timing experiments are two different assays with different sensitivities. The authors need to verify that replication timing assay can in fact pick up on intermediate phenotypes with a similar sensitivity as the Hi-C.

A direct comparison would be impossible. However, the replication timing method has certainly been demonstrated to pick up differences that are much smaller and localised than in the cases analysed here. For example, in the case of Brg1 and Baf250a knock outs, the sensitivity was able to reveal differences between the samples as small as 10% (Takebayashi et al., Epi & Chrom. 6, 2013). A comment was added in the discussion to highlight this point.

Minor points

1. Could the authors specify if there are any known defects to the deletion of the PP1 interaction sites of Rif1? Can this deletion potentially impact other interactions of Rif1? (It seems it doesn't impact DNA binding but how about multimerization?). Please discuss briefly in the article.

This is an interesting question. We do not have a reliable measure of multimerization in vivo. However, an indication that Rif1- Δ PP1 does not have at least a dominant-negative type of effect on RIF1 function (as you would possibly expect from a multimerization mutant) comes from the Rif1- Δ PP1 prior to deletion of the floxed allele. If the mutation were to affect multimerization, we should have detected some defect. Instead, at least cell cycle distribution and MERVL expression are as seen for the wild type, prior to tamoxifen-induced hemizygoty.

2. Figure 1F: Could the authors please represent this figure as absolute amounts of MERVL RNA normalized to the house keeping gene and include the individual data points? The retrovirus DNA is usually repressed in cells, thus very little MERVL RNA would be present in WT cells; 20 fold more of almost nothing, is, still almost nothing. If this is the case, it would not be appropriate to say there is significant gene expression changes or that MERVL is de-repressed.

This has been modified according to the suggestions and moved to Fig. S2B, according to the indication of the Reviewer 3, point 6.

3. Please clarify and explain this sentence: " The replication profiles of both Rif1-DPP1 and Rif1-KO mESCs appear similarly compressed around the zero (Fig. 2B and S2B)...".

This is one example where the writing could be improved to appeal to a wider audience. I think the authors findings have implications beyond the field of DNA replication and thus the article will benefit from appealing to a wider audience who are not familiar with specific assays.

We have modified the sentence and in other similar places in the text.

4. Figure 4: Perhaps best to label Figures A, B, C, D and E in order.

We have changed the order of the panels as suggested.

5. What is the y-axis scale in Fig S5B? Is it log2?
Can the authors present S5B and C in linear scale?

This data is now in fig. S6B and C. We have changed the plots to a linear scale.

Reviewer #3 (Remarks to the Author):

Rif1 is a large polypeptide that has been linked to the control of DNA replication timing, double strand break repair and chromatin organization in different experimental systems. Previous studies have shown that at least part of Rif1 functions require its interaction with protein phosphatase 1 (PP1), which counteracts the phosphorylation of MCM proteins at late replication origins. This new manuscript from the Buonomo lab reports a set of experiments to address how chromatin organization and replication timing (RT) are affected by changes in Rif1 genetic dosage or by the loss of the interaction between Rif1 and PP1.

The authors have genetically modified mouse embryonic stem cell lines to knock-in either WT Rif1 or a mutant version designed to disrupt its interaction with PP1. This is done in a cellular system in which the second Rif1 allele is conditionally inactivated by Cre-Lox (Figures 1 and S1). They show that chromosomal RT profiles are affected in Rif1- Δ PP1 cells almost as much as in Rif1 KO cells, but not by having Rif1 in hemizyosity (Figures 2 and S2-S3). The spatial distribution of replication foci is inferred by the analyses of patterns of EdU foci and other markers (Fig S4), as well as the number and clustering of replication forks by 3D-structure illumination microscopy (Fig. S5). Finally, chromatin organization Hi-C assays (Figures 3-4) reveal that the loss of Rif1-PP1 interaction alters chromatin contacts as much as Rif1 KO cells. From this combination of approaches, the authors propose that both nuclear architecture and RT depend on the Rif1-PP1 interaction, and that these two aspects can be partially uncoupled.

The lack of a strict correlation between 3D chromatin contacts and RT is interesting and in line with the recent findings that alterations in cohesin or CTCF have little or no impact on RT. In my opinion, this study provides a moderate advance in our understanding of the co-regulation and causal links between chromatin organization, RT, and transcriptional programs. The bigger mechanistic question at this stage would be the molecular target(s) of Rif1-PP1 that regulate chromatin contacts and nuclear compartmentalization.

General comments:

1. Some of the Results are a bit difficult to follow. A possible reason is that the study is apparently focused on the Rif1- Δ PP1 mutant version, which mimics many of the phenotypes caused the complete ablation of Rif1 and confirms the importance of Rif1-PP1 interaction in mESCs. But the conclusion that 3D chromatin contacts and spatial organization of DNA replication factories can be separated from RT is largely derived from the analyses of WT Rif1 in hemizyosity (sufficient to change chromatin conformation without affecting RT). My impression is that the information does not always flow naturally, and this may be accentuated in some cases by the figure design (see below).

We have modified the text, changing the order of presentation where suggested and tried to eliminate the technical jargon wherever possible.

2. The experiments showing alterations in the patterns of replication foci (Figure S4; quantification pie plots in Figure 2D) are inherently complicated because changes in Rif1 levels alter the markers normally used to call "early S" or "late S" patterns. A related limitation occurs in the analyses of fork activity by SIM, as acknowledged in the text (Fig S5; "Since Rif1 deletion and expression of Rif1-DPP1 induce a loss of equivalence between replication foci distribution and replication timing, we could not analyse early, mid and late S phase separately"). In both cases, the results would be more conclusive if the IF (Figure S4) and SIM experiments (Fig S5) were performed in cells synchronized and collected at different time points along S phase. A simple possibility would be to sort cells by FACS based on their DNA content: regardless of the duration of S phase in each case, three cell fractions sorted with increasing amounts of DNA (between 1C and 2C) would necessarily correspond to early, mid and late S phase.

We have considered this, however, the shape of the cells after FACS sorting and cytopsin or short adhesion to attach cells to the slides, is badly altered, making the acquisition of 3D images of sufficient quality impossible. Instead, we have used DAPI quantification to assign the cells to early or late S-phase, as published in Heinz et al., 2018, NAR 46, 6112-6128. The message has not really changed as compared with the non sub-staged cells. We have included the new data in Fig. S6B and C.

3. The Hi-C results should be presented and discussed with more clarity for the non-specialist. For instance, the results with Rif1- Δ PP1 (Figure 3A-B) are not mentioned initially (page 8, last paragraph) but are discussed shortly afterwards, creating a couple of confusing sentences.

We can appreciate that this made the flow harder to follow. The text has been changed according to the suggestion.

Also, in the saddle plots in Figure 4A, loci are ranked by their replication timing. Therefore, compartmentalization is not directly derived from this analysis (Figure legend reads "A and B indicates the compartments"), but rather inferred from previous studies, correct?

For Fig. 4A, the reviewer is correct, the plots are based on RT, so A and B are labelled based on the previously known relationship between compartments and RT. In Fig. 4E the compartments were re-calculated from our data. We added a clarifying sentence in the figure legend.

Specific suggestions about the Figures:

4. In Figure 1A-B, the expression levels of HA-Rif1 in the different cell lines are shown in a rather unusual manner (distribution of intracellular flow cytometry staining with anti-HA, in arbitrary units). What would be the background signal in this assay, e.g. without secondary antibody? Why not include also a simple immunoblot for the different clones (as in the example shown in Figure 1C)?

Fig. S1C shows the immunoblot for the same samples as Fig. 1A-B. We have added the FACS analysis to be more quantitative. Please see the file attached below, with all the controls: 15_10= Rif1-KO; A7, H4, H6= Rif1-TgWT; H1, H2, G11=Rif1-DPP1; FH11=^{FH}/FH Rif1; RFHF14=^{+FH}Rif1; Alexa=only click reaction, no DAPI; GFP_DAPI=no click reaction, only DAPI; NS= no staining; GFP= control if any GFP signal from Oct4-GFP reporter present in the cells is left after fixation and interferes with the far red channel; GFP_Alexa= click reaction, and GFP channel, for Oct4-GFP.

5. Figure 1D and Supp 1D contain histograms showing two replicates of the same experiment. I would recommend: (a) showing the flow cytometry plots for a representative experiment;

A new Suppl. Fig. has been added with an example of the data (Fig. S2A)

in this case, a control with another protein whose chromatin association fluctuates in the cell cycle (e.g. one of the MCM proteins) could be shown to validate the technical approach;

This technique has been used in many papers (for example, for MCM3 and ORC1 in Hiraga et al. EMBO Rep. 2017). In fact, it was first published years ago for human MCM proteins by Bruce Stillman, in the paper cited in the text. However, intra-cellular FACS only works with some combination of protein/antibody. We would need to test several of the commercially available ORC1 or MCM3 antibodies to find one that works for this application AND in mouse cells (which is generally the major problem). Since the conclusions from this experiment are not dependent upon absolute amounts, we feel that they are valid without this additional positive control.

(b) showing the histograms with the quantification and statistics of three replicates.

The data have been combined in one histogram and statistics are shown (Fig. 1D).

6. Figure 1F is not very informative (esp. given the large variation between replicates) and a bit separate from the rest of the study. It could be shown as Supplementary material.

We agree with the Reviewer, the panel has been moved in Fig. S2B. We have also modified the plots, following the suggestion of Reviewer 2, point 2.

7. Figure 2B could benefit from a change in colors, particularly for Rif1-WT and Rif1-FH lines (very similar in the current version).

Rif-WT is now darker grey. Hopefully this helped distinguish them.

8. In current Figure S4 or its future equivalent, showing examples of Rif1-TgWT and Rif1- Δ PP1 nuclei would improve the message.

The reason this was not shown is that the Rif1-FH cells used to generate Rif1- Δ PP1 and Rif1-TgWT express a fluorescent transgene under the control of Oct4 promoter, used during embryonic stem cells derivation to monitor

pluripotency. This impedes the 4-colour staining, as we have done in the Rif1-WT and Rif1-KO cells. To circumvent this issue, in the new fig. S5, we have added a panel (new panel B) where we show EdU patterns in FACS sorted cells of every genotype.

REVIEWERS' COMMENTS

Reviewer #2 (Remarks to the Author):

With the exception of a few points where research material/tools are limiting, the authors have satisfactorily addressed my concerns; I would recommend the publication of this article. It is timely, interesting and can appeal to a wide audience.

I will be looking forward to read this article at Nature Comm soon.

Reviewer #3 (Remarks to the Author):

The authors have addressed most of my comments to the previous version. In particular, they now have included analyses of replication foci and fork activity in cells classified as early, mid and late S phase according to the intensity of DAPI staining (new Supp Fig 6B,C), based on a published method. The actual DAPI quantification details are not included in the new Figure, but I assume that the interpretation of the authors is correct.

In addition, they have modified the text in several sections for the sake of clarity. I believe the manuscript reads better now and is suitable for publication.

Juan Mendez -CNIO, Madrid.